# Wearable-Based Stair Climb Power Estimation and Activity Classification

**DOI:** 10.3390/s22176600

**Published:** 2022-09-01

**Authors:** Dimitrios J. Psaltos, Fahimeh Mamashli, Tomasz Adamusiak, Charmaine Demanuele, Mar Santamaria, Matthew D. Czech

**Affiliations:** Pfizer Inc., 610 Main Street, Cambridge, MA 02139, USA

**Keywords:** stair climb power, accelerometer, gyroscope, inertial measurement units, machine learning, remote monitoring, gait

## Abstract

Stair climb power (SCP) is a clinical measure of leg muscular function assessed in-clinic via the Stair Climb Power Test (SCPT). This method is subject to human error and cannot provide continuous remote monitoring. Continuous monitoring using wearable sensors may provide a more comprehensive assessment of lower-limb muscular function. In this work, we propose an algorithm to classify stair climbing periods and estimate SCP from a lower-back worn accelerometer, which strongly agrees with the clinical standard (r = 0.92, *p* < 0.001; ICC = 0.90, [0.82, 0.94]). Data were collected in-lab from healthy adults (*n* = 65) performing the four-step SCPT and a walking assessment while instrumented (accelerometer + gyroscope), which allowed us to investigate tradeoffs between sensor modalities. Using two classifiers, we were able to identify periods of stair ascent with >89% accuracy [sensitivity = >0.89, specificity = >0.90] using two ensemble machine learning algorithms, trained on accelerometer signal features. Minimal changes in model performances were observed using the gyroscope alone (±0–6% accuracy) versus the accelerometer model. While we observed a slight increase in accuracy when combining gyroscope and accelerometer (about +3–6% accuracy), this is tolerable to preserve battery life in the at-home environment. This work is impactful as it shows potential for an accelerometer-based at-home assessment of SCP.

## 1. Introduction

The use of digital technology in medicine can improve patient monitoring, diagnosis, and treatment through the use of novel devices and algorithms. While clinical assessments provide only a single snapshot in time, digital devices offer the opportunity for continuous monitoring of physical function and physiological measures [1,2,3]. To this end, inertial measurement unit (IMU) devices, generally consisting of both a triaxial accelerometer and gyroscope, have been used extensively to quantify motor function and quality of life in the home environment [4,5]. Specifically, IMU-based algorithms have been developed and validated for measuring various clinically relevant metrics, including physical activity, gait, and sleep [2,6,7,8,9,10,11,12].

One important aspect of physical mobility currently lacking a digital measurement solution is stair climbing performance. Muscle strength is a valuable clinical measure to evaluate disease state and treatment effect [13,14]. By extension, leg muscular strength is a health attribute related to falls, balance, and functional independence in community-dwelling older adults [15]. In order to measure leg muscular strength, conventional exercise equipment requiring trained personnel is often used. The Stair Climb Power Test (SCPT) was developed to estimate leg muscular strength and has become a standard clinical assessment commonly applied to various disease populations, including cachexia [16], osteoarthritis [15,17], diabetes [15], Parkinson’s disease [18], and cardiopulmonary disorders [15,16,17,18,19]. Stair climb power (SCP), the clinical metric estimated from the SCPT, is a function of muscular work per unit of time, and only requires a stopwatch, knowledge of the participant’s weight, and a staircase to estimate [19]. The SCPT was originally validated on a 10–11 step staircase, however, since a large staircase is not feasible in many clinical settings, the test has been adapted and validated to a four-step staircase. Prior work has established a functional response threshold of ≥10% increase in stair climb power after 12 weeks of selective androgen receptor modulator treatment in non-small cell lung cancer patients [16]. A decrease in stair climb power, in elderly populations, has been associated with changes in balance, falls, and morbidity and mortality, while increases have shown an improvement in quality of life [16]. Additionally, stair climbing performance has been shown to positively correlate with cardiorespiratory efficiency, VO_2_ max, and lower limb strength [20].

Despite the SCPT’s clinically relevant benefits for quantifying muscle strength and mobility performance, the clinical assessment has drawbacks. The SCPT requires a trained clinician to manually record time on a stopwatch, introducing an element of human error, and can only be conducted periodically in-clinic [21]. Therefore, there is a need for a low burden, quantitative, and continuous assessment of leg muscle strength. The SCPT is frequently assessed due to its association with activities of everyday living [22]. In fact, due to the repetitive nature of the SCPT, and the test’s application in everyday living, the SCPT is well suited for an at-home assessment, provided that the participant can climb stairs [21]. In this case, wearable technologies may provide a feasible solution for at-home evaluation of the SCPT.

Estimation of stair climb performance in the at-home environment requires both the continuous detection of stair ascending events and estimation of stair climb power. Prior works have attempted to identify stair walking events using IMUs positioned at various locations, including waist, shank, chest, wrist, foot, and demonstrated varying degrees of accuracy [17,23,24]. Several studies have investigated the classification of stair climbing events from a lumbar worn IMU [21,25,26,27,28,29]. A lumbar worn IMU is minimally obtrusive and suitable for remote patient monitoring of physical activity, such as gait and sit-to-stand, in various disease populations [21,30,31]. It is unclear whether both a gyroscope and accelerometer positioned at the lumbar are needed to detect ascending stair climbing in an at-home environment. The added requirement of a gyroscope significantly increases data size and shortens device battery life, and thus requires frequent data transfers and charging, impacting patient burden and compliance. Furthermore, there have been limited attempts and a lack of publicly accessible methods for estimating stair climb power from periods of stair ascent [32].

To address current shortcomings, in this work we (a) present StairPy, a sensor-based software package for estimating stair climb power from a lumbar-positioned accelerometer and (b) investigate the tradeoffs of sensor modality and classifier for distinguishing stair ascending from stair descending, and stair walking from level walking (gait). Currently, there are no end-to-end methods for stair climb detection and estimation of stair climb power using wearable sensors. The method proposed in this work demonstrates feasibility for sensor-based stair climbing performance measurement.

### 1.1. Materials and Methods

#### 1.1.1. Participants and Procedure

Data were collected in a Pfizer Inc. internal study, enrolling healthy volunteers [11]. Study participants (*n* = 65) covered two age groups: younger [18–40 yo, *n* = 33] and older [65–85 yo, *n* = 32]. Participants were equally distributed by sex in the older and younger age groups. The mean age for the younger group was 29.2 ± 4.54 yo and older group 72.3 ± 5.51 yo. The mean BMI for the younger group was 23.4 ± 2.62 and 24.5 ± 2.60 for the older group. The distribution of subject height and weight are shown in Appendix A. Sixty-four participants were used for analysis as one participant had a sensor recording error. These participants were recruited from the surrounding area of Cambridge, MA, USA.

Inclusion criteria included male or female participants aged 18–40 or 65–85 yo inclusive, who had no significant health problems as reviwed by a certified clinician at intake. Additionally, participants were required to have a body mass index (BMI) between 18.5 kg/m^2^ and 30 kg/m^2^, or an absolute weight < 125 kg. Lastly, the participants had to be able to read, understand, and provide informed consent. This was evidenced through a personally signed and dated informed consent document indicating that the participant has been informed of all pertinent aspects of the study.

Exclusion criteria included participants who self-report any medical condition, recreational substance use, or medication use which would prevent them from completing study tasks or impair the providing of informed consent. Additionally, any participants with a VES13 (Vulnerable Elders Survey) total score > 3, including a 0 in all ADLs (Activities of Daily Living), were excluded from the study. Furthermore, participants with skin nickel, scilicone or adhesive allergies, or any participants with implanted devices (pacemakers, electric pumps, etc.) were excluded from the study. Lastly, any participants with relationships to the study staff or clinicians were excluded from the study.

Each participant had two in-clinic visits, where they performed a battery of tasks including the Short Physical Performance Battery (SPPB) [33] and SCPT, while instrumented with a six sensor IMU set (OPALs by APDM, Inc.—Mobility Lab Software v2.0.0.201903301644 ERT Technologies (Portland, OR, USA)) [34]. The APDM OPAL sensors were placed on the participant’s sternum, lower back (lumbar), both wrists and both feet. These sensors contain a tri-axial accelerometer and gyroscope, which collect data at 128 Hz. Data were collected and stored on each sensor, and then locally downloaded using the Mobility Lab software, and exported as deidentified H5 files.

While instrumented, participants were instructed to climb a four-step staircase at their natural pace, using the handrails as necessary. Once both feet reached the top step, participants were instructed to pause, then turn around and descend the stairs. A trained technician timed the ascending portion using a stopwatch, from saying “go” until the participant’s second foot touched the top step. The SCPT was completed three times with a 30 s break in between each trial in order to reduce fatigue.

Subjects also performed the SPPB during their in-clinic visits [33]. The SPPB included a gait task, which required participants to walk three laps on a 6 m (~20 feet) GAITRite mat [35] in one recording, at a comfortable pace. These gait tasks were used to produce negative training class data.

#### 1.1.2. Feature Extraction

Video from the in-clinic visit was used to annotate periods of stair ascending and descending, for each SCPT performed. Annotations were used to segment the time-series signal from the lumbar position IMU into stair climb ascending and descending events. Periods of ascending and descending events, containing time-series triaxial-accelerometer and triaxial-gyroscope data, were segmented into 1.5 s windows with 50% overlap, similar to the method used in Hellmers et al. [21]. These segments were then downsampled to 50 Hz in order to make the results generalizable to other sensors with lower sampling frequencies. The same method was used to extract 1.5 s segments from a structured level walking task, which were used as a negative training class.

Signal features were derived from each 1.5 s segment of stair ascent, stair descent, and gait activity. The signal features derived included a combination of features from previous literature, including signal mean, min, max, variance, root mean square, standard deviation, entropy, autocorrelation, correlation coefficient, and dominant frequency [21,25,36]. Each feature was derived from the three accelerometer and gyroscope axes. Several features were also derived from accelerometry data, including signal range and wavelet derived measures, noted by Weiss et al. (2016) as being important for distinguishing stair climbing from gait [25]. This feature extraction process can be found in the code for the StairPy module in Appendix A.

#### 1.1.3. Stair Climbing Classification

Two machine learning binary classifiers were trained; one to distinguish stair walking from gait events, and one to distinguish stair ascent and descent. To determine the best algorithm for the training, four supervised machine learning algorithms were trained (Python 3.7, Scikit-learn (0.24.2)) for each classifier: K-Nearest Neighbors (Knn), Random Forest (RF), Logistic Regression (LogReg), and an Ensemble Voting Classifier (Ensemble) which used a combination of these three models [37]. Knn, RF, and LogReg are widely used supervised machine learning models, are easily trained, and commonly used for classification problems [38,39,40]. The Ensemble model leverages the outputs of all three classifiers to make predictions. Separate models were trained on only accelerometer features, only gyroscope features, and a combined accelerometer and gyroscope feature set to compare tradeoffs and synergy between sensors. A total of 148 features were generated for each 1.5 s segment; 72 gyroscope features and 76 accelerometer features (Appendix A). The dataset used for the stair walking vs. gait classifier contained 4808 total observations (positive and negative classes combined) and the stair ascending vs. descending classifier dataset contained 2384 total observations.

Positive and negative classes were balanced (1:1) across subjects per visit and task. This was carried out by combining the number of stair ascending and descending segments to be used as a positive class for a particular task, and then randomly selecting the same number of gait segments (performed during the same visit, by the same subject) to be used as a negative class. A similar method was used to balance the stair ascending vs. descending classifier, where the positive and negative classes were stair ascending and descending, respectively. Features with correlation values > 90% were removed first, and the remaining features were further reduced using Recursive Feature Elimination (Scikit-learn—RFECV, version 0.24.2) [37,41]. Ten-fold cross-validation was performed on the final featureset to determine overall model performance. Subjects were partitioned 80% and 20% for training and testing respectively. During each round of 10-fold cross-validation, a grid search approach was used to determine the best hyperparameters for each model. Sklearn feature importances were used to assess the importance of model features and their impact on classification [37].

#### 1.1.4. Stair Climb Power Calculation (StairPy)

Raw accelerometer segments from stair ascending events were used to develop a step detection algorithm for stair climbing (StairPy) (Appendix A). Only the acceleration in the vertical direction was used to detect steps. The raw vertical acceleration signal was first transformed using a Continuous Wavelet Transformation (CWT) function to detrend the data. Next, the signal was filtered using a low-pass butterworth filter and the output was integrated using the composite trapezoidal rule. Finally, a gaussian wavelet transform was applied to the signal and the resulting transformed signal underwent peak detection, and the corresponding peaks were identified as steps. This method was adapted from previous CWT based algorithms, using the vertical axis of a lumbar worn accelerometer, to identify gait step times [7,8,11]. The height of each step on the staircase used in this study was 0.154 m [42]. The difference in time between each step (step time) was calculated and used in power estimation. Standard stair climb power computation follows Equation (1) in which *Power_sc_* is the stair climb power in Watts, *m* is the participant’s mass in kg, *g* is the acceleration due to gravity (9.81 m/s^2^), *h* is the height climbed in meters, and *t* is time in seconds to climb the height *h*:(1)$$Power_{sc}=\frac{mgh}{t},\;h=\mathrm{height}\mathrm{of}\mathrm{staircase}\;(\sim 0.6\;m)$$

More specifically, StairPy power calculation follows Equation (2) in which details of time and height calculation are clarified:(2)$${Power}_{sc}=\frac{mg\left(\frac{h}{4}*steps\;detected\right)}{t_{0}+\left(t_{2}-t_{1}\right)+\left(t_{3}-t_{2}\right)+\left(t_{4}-t_{3}\right)},\;\frac{h}{4}=step\;height\;(\sim 0.15\;m),\;t_{0}=mean(t_{2}-t_{1},\;t_{3}-t_{2},\;t_{4}-t_{3})$$

In this equation, the sum of the step heights $\frac{h}{4}$ and step times $\left(tn-t_{n-1},\;n=2,3,4\right)$ for each detected step by the algorithm are used. Equation (2) includes a term (*t*_0_) for the time it takes the participant to reach the first step, a time which is included in the clinical assessment but not able to be detected by our algorithm. This *t*_0_ term is estimated as the mean of detected step times (*mean*(*t*_2_ − *t*_1_, *t*_3_ − *t*_2_, *t*_4_ − *t*_3_)) and accounts for the time to first step.

Figure 1 outlines the differences in stair climb power estimation between the Standard (four-step) SCP estimation (Figure 1A) and StairPy SCP estimation (Figure 1B). Standard SCP is estimated using the mass of the participant, gravity, total height of the staircase, and the time it takes for the participant to reach the top of the stairs. Similarly, StairPy calculates stair climb power as a function of number of detected step times (time between alternating initial contacts), height of each step, the sum of detected step times, mass of the participant, and gravity (Figure 1B).

To reduce misidentified steps derived from the step detection algorithm, the distribution of stair climb step times was visualized across participants. Given the four-step staircase used in the study, we expected four foot-contacts to be detected by the algorithm, assuming all participants alternated steps. All step times exceeding 1200 ms (1.2 s) corresponded with misdetection of steps (Figure 2A,B). Therefore, a threshold of 1.2 s was used to reduce misidentified steps. An example of a task with correctly identified steps can be seen in Figure 3A. All the time intervals between detected steps were below 1200 ms. In contrast, Figure 3B shows failure to detect the second stair climb step (next detected step is the 3rd step). In this case, we would not use the time interval between the first and second detected step because it was above 1200 ms. We would use the time interval between the 2nd and 3rd detected steps and use that for power computation. StairPy was able to correctly identify four initial contacts in 92.9% of trials (*n* = 378).

### 1.2. Statistical Analyses

Statistical analysis was performed in R version 4.0.4 with following main packages: “BlandAltmanLeh” for Bland–Altman plots, and “irr” for intraclass correlation coefficient (ICC). ICC is a statistical measure of reliability, measuring not only the correlation between measures, but also their absolute agreement [43]. The mean SCP values across trials and visits were used for statistical analysis. A Bland–Altman plot and 95% limits of agreement were used to evaluate the agreement between StairPy and clinically-derived SCP values. Agreement between the two SCP values was further characterized with ICC_2,1_ (two-way random effects, absolute agreement, with respect to single measurement) with lower and upper confidence bounds, reported as ICC (lower, upper). Pearson’s correlation coefficient was also computed to test for the consistency in the SCP estimation by the two methods. Group comparisons to test of age and sex effects were performed using 2-sample *t*-tests.

## 2. Results

### 2.1. Stair Climb Power Can Be Accurately Measured Using a Lumbar-Worn Accelerometer

The calculated stair climb power using StairPy showed a strong relationship with the Standard method (Pearson’s *r* = 0.92, *p* < 0.001) (Figure 4A). Furthermore, using ICC analysis, StairPy showed good agreement compared to the gold standard (ICC = 0.90, [0.82, 0.94]), with a systematic bias of −4.11 watts (Figure 4B). There was no significant difference between younger and older cohorts for both power methods (Standard SCP: *t*(57.99) = 0.05, *p* = 0.96; StairPy SCP: *t*(56.37) = −0.87, *p* = 0.39); Figure 5A. Male participants exhibited higher stair climb power than female participants and the effect was similar for both power estimation methods (Standard SCP: *t*(56.84) = 3.22, *p* = 0.002; StairPy SCP: *t*(61.75) = 3.21, *p* = 0.002); Figure 5B.

### 2.2. A Lumbar-Mounted Accelerometer Can Distinguish Stair Walking from Gait, and Stair Ascending from Descending

A particularly challenging problem for identifying periods of stair walking using an IMU is distinguishing stair walking from periods of gait, due to the similarity of the two physical activities [44]. Therefore, to evaluate classification performance for detecting stair walking events, machine learning models were trained using periods of stair walking and gait as positive and negative classes, respectively. Using various accelerometry signal features, deemed important for stair walking classification from prior literature [21,25], our optimal model for distinguishing stair walking and gait was an Ensemble model which achieved an accuracy of 91% [sensitivity = 0.91, specificity = 0.92]. The RF model also achieved high accuracy, 89% [sensitivity = 0.90, specificity = 0.89]. A Knn model achieved a slightly lower accuracy of 87% [sensitivity = 0.85, specificity = 0.89], and likewise a LogReg model achieved a slightly lower accuracy of 88% [sensitivity = 0.89, specificity = 0.88].

Since the ascending portion of stair walking is typically used to measure leg strength, a second group of models were trained to distinguish stair ascending from descending. Classifying stair ascending and descending was achievable using the Ensemble model, which achieved an accuracy of 89% [sensitivity = 0.89, specificity = 0.90]. The RF binary classifier achieved a slightly lower accuracy of 87% [sensitivity = 0.87, specificity = 0.88]; similar to the LogReg model which achieved an accuracy of 88% ([sensitivity = 0.87 specificity = 0.89], and the Knn model which achieved a slightly lower accuracy of 88% [sensitivity = 0.89, specificity = 0.87].

### 2.3. Tolerable Difference in Classification Performance Using Gyroscope or Accelerometer Signal Features Alone

To evaluate the tradeoff between using gyroscope or accelerometer features for classifying stair walking events from gait, accelerometer feature-based model results were compared to model results using gyroscope features alone. Compared to the accelerometer based models, model performance decreased for all models when using gyroscope features alone to distinguish gait and stair walking (Figure 6A). The RF model decreased in accuracy by 5%, the LogReg model decreased by 6%, the Knn model decreased by 3%, and the Ensemble model decreased 6% in accuracy when using gyroscope features only (Table 1).

Minimal differences were seen between using the accelerometer or gyroscope-based features alone to distinguish between stair ascent and descent (Figure 6B). Compared to the accelerometer based model, there were minimal differences when training models on gyroscope features alone. The RF model accuracy inceased in accuracy by 1%, the Knn model decreased by 2%, the LogReg model decreased by 5%, and the Ensemble model decreased by 1% in accuracy when using gyroscope features alone (Table 2).

### 2.4. Classification Performance Is Slightly Enhanced Using Gyroscope and Accelerometer Together

The stair walking vs. gait classifier accuracy was slightly improved when gyroscope and accelerometer features were combined (Figure 6A). Compared to the accelerometer feature based RF model, the accelerometer and gyroscope feature RF model increased accuracy by 3%, the Knn model increased accuracy by 4%, the LogReg model increased accuracy by 4% (Table 1), and the Ensemble model increased in accuracy by 3%.

Similarly, stair ascending and descending classification accuracy was improved using a combination of gyroscope and accelerometer signal features (Figure 6B). The RF model accuracy was increased by 7%, the LogReg model accuracy was increased by 5%, the Knn model accuracy increased 6% (Table 2), and the Ensemble model increased in accuracy by 6%.

### 2.5. Feature Investigation

Figure 7A shows feature importance values for the top 20 accelerometer features based on their impact on the ensemble stair walking vs. gait classifier. Several predominant features for stair walking event classification were observed, including: accelerometer x autocorrelation, accelerometer x maximum, accelerometer x standard deviation, accelerometer z median, accelerometer z skewness, accelerometer y autocorrelation, and accelerometer y 20th percentile.

On the contrary, Figure 7B shows feature importance values for the top 20 features based on their importance for the ensemble ascending vs. descending classifier, trained on accelerometer features. In this model, classification was dominated by several predominant features: accelerometer xz correlation coefficient, accelerometer x FFT energy, accelerometer z minimum, accelerometer z standard deviation, accelerometer x mean, accelerometer x maximum, and accelerometer z median.

## 3. Discussion

### 3.1. Main Findings

This work is significant as it: 1. Presents a novel open source python package for the estimation of stair climb power, 2. Compares different sensing modalities for the classification of stair walking and stair ascent vs. level walking and stair descent respectively, and 3. Demonstrates the ability for a ensemble classifier to exceed 89% accuracy for stair walking and stair ascent classification. In this work, we developed and tested a method to estimate stair climb power from a lumbar worn accelerometer that has strong agreement (Pearson’s r = 0.92, *p* < 0.001, ICC = 0.90, [0.82, 0.94]) with the traditional clinical assessment. Sensor-based stair climb power estimation showed similar age and sex stratified results compared to the clinical assessment. In addition, we demonstrate the ability to classify ascending stair climbing using an accelerometer sensor. Our results suggest there is minimal tradeoff in model performance between using accelerometer or gyroscope sensors alone. Specifically, for distinguishing between stair walking and gait, the use of accelerometer features alone resulted in an accuracy of 91%, only 3% lower than the combined accelerometer and gyroscope model. Similarly, for distinguishing between stair ascending and descending, using accelerometer or gyroscope features alone yielded an accuracy of 89%, which was only 6% lower than the combined accelerometer and gyroscope feature trained model. The slight difference in classification accuracy between the sensor combinations suggests accelerometry alone may be sufficient to classify stair ascending events, although slightly superior accuracy can be achieved by adding a gyroscope signal. However, there is a slight benefit to using both accelerometer and gyroscope signals combined, this improvement is tolerable compared to the accelerometer model. In the at-home environment, maximizing battery life is critical. Increasing the number of sensors (gyroscope plus accelerometer) has worse battery life compared to an accelerometer alone. With this in mind, a slight decrease in classification accuracy to maximize battery life is a worthy tradeoff. Our results support the feasibility for sensor-based stair climbing assessment, which could be expanded to monitoring of free-living physical activity in an at-home environment.

These results are relatively consistent with prior works using a waist worn accelerometer to classify stair walking events. Hellmers et al. was able to distinguish stair walking from gait and ascending from descending stairs with about 86–90% accuracy, and estimated stair climb power with a mean deviation of 2.35% compared to the clinical assessment [21]. Weiss et al. was also able to achieve about 94% accuracy when distinguish stair ascent from stair descent and gait using an accelerometer worn on the waist [25]. While our model achieves a slightly lower accuracy, it is important to note that the models used in these previous works are performed on 10- to 12-step staircases, which may allow for more stair climbing periods for training data, and only used healthy elderly participants.

### 3.2. Accuracy of Stair Climb Power from Acceleration Data

The stair climb power package (StairPy) presented in this paper shows a strong relationship with the gold standard, clinical timed assessment (r = 0.92, *p* < 0.001, ICC = 0.90, [0.82, 0.94]). However, the algorithm minimally overestimated stair climb power (–4.11 watts), as shown in Figure 4B following application of a correction factor. This bias is minimal and irrelevant in a clinical trial context. In clinical trials, change over time is needed to express effect. With this in mind, this bias is inconsequential as long as the patient shows change over time from baseline.

The correction factor was applied due to inherent differences between the accelerometer-based method and the clinical stopwatch approach. This mean difference between StairPy and the clinical stair climb power assessment is due to variation in how power is estimated between the two methods (Figure 1). For example, the clinical assessment is timed from “go” until both feet touch the top stair, which includes the period between “go” and the participants first step on the staircase. However, the period between “go” and the first step does not include body movement in the vertical direction, and thus leads to an underestimation of vertical muscular work and stair climbing power. To account for this difference, StairPy estimates the added time between the moment the clinician says “go” and the participant’s first step (*t*_0_) on the staircase as the mean of the detected step times and includes this in the total stair climbing duration. Appendix A shows the average first step time (*t*_0_) across subjects vs. ICC score, a measure of agreement between StairPy SCP and the clinical SCP. The average first step time across all subjects was 0.64 s, which achieves a high ICC score and a reliable estimation of SCP, as shown in Appendix A.

In the context of studies conducted outside of the laboratory, it may be reasonable to remove this bias correction step. The uncorrected stair climb power estimate is likely a more accurate estimate of true stair climb power and the effect of a consistent bias may be inconsequential in the context of clinical studies, as stair climb power change over time would likely not be affected.

### 3.3. Tradeoffs between Sensor Modalities for Classifying Stair Climb Events

Classifying periods of stair climbing behavior is necessary for evaluating stair climb performance. To this end, we observe limited tradeoff between using accelerometer or gyroscope signals alone for distinguishing both stair walking from gait and stair ascending from descending. Furthermore, we find that a combination of accelerometer and gyroscope signals provided slightly improved ability (~3–6% improvement in accuracy) to distinguish both stair walking from gait and stair ascending from descending, compared to accelerometer or gyroscope signals alone. This is understandable given the sensors ability to not only measure acceleration during stair walking, but also the rotation of the torso. While, a 3–6% improvement in classification accuracy is desirable, our work shows the feasibility of using an accelerometer alone for stair ascent classification. The addition of gyroscope and other sensors, such as barometers or magnetometers, may improve results in detecting stair climbing [17,45,46], but commercially available devices that include additional sensors are limited, significantly reduce battery life, and thus may not be as well suited for long-term continuous at-home assessment. Our findings suggest that accelerometers can provide a solution for sensor-based classification of stair climbing and assessment of free-living activity in at-home environments.

### 3.4. Discussion of Features

For the stair walking versus gait classifier, we chose to include level walking as the sole negative class and stair walking as the positive class due to the challenge of distinguishing the two similar movements. Even so, there are indeed slight differences between level walking and stair walking, particularly in the movement of the hips and trunk, where our lumbar sensor was instrumented in this study.

During level walking, the gait cycle starts with an extended knee followed by a continued stance and swing phase, with limited hip flexion. On the contrary, during stair walking, there is a 50–60° flexion of the hip to start the gait cycle, followed by decreased hip flexion (10–15°) during the weight acceptance phase as the participant ascends to the next step [21,47]. Additionally, approximately 10–20° of additional range of motion from the lower limb joints is required for stair walking compared to level walking [21]. Stair walking also requires more trunk stability and lower limb muscle activity in order to propel the participant up the staircase.

Slight differences in lower limb biomechanical movements between gait and stair walking were reflected in our results. For example, distinguishing stair walking from gait relied on a combination of accelerometer signal features from the vertical (x), anterior–posterior (z) and medial-lateral (y) axes. The features we observed that best captured differences in trunk movement between stair walking and level walking were statistical features (accelerometer x autocorrelation, accelerometer x maximum, accelerometer x standard deviation, accelerometer z median, accelerometer z skewness, accelerometer y autocorrelation, and accelerometer y 20th percentile). These features likely capture the vertical movement and additional trunk flexion required as participants shift weight to transfer their foot to the next step when ascending or descending stairs.

Additionally, the most important features for stair ascending vs. descending classification were centered around the vertical (x) and the anterior–posterior (z). The vertical axis captures the acceleration up or down during stair ascending or descending. The anterior–posterior axis is also a major component of stair climbing as the torso bends along the anterior–posterior axis with limited medial-lateral movement. The most important features for distinguishing stair ascending and descending were statistical features (accelerometer xz correlation coefficient, accelerometer z minimum, accelerometer z standard deviation, accelerometer x mean, accelerometer x maximum, and accelerometer z median), and frequency domain features (accelerometer x FFT energy), all of which were derived from the accelerometer’s x and z axes. These features likely capture the vertical and anterior–posterior movement of the torso as participants ascend or descend the staircase.

Our results show that a combination of all three accelerometer axes is important for distinguishing stair walking and gait, though only the vertical and anterior–posterior axes were most important when distinguishing between stair ascent and descent.

### 3.5. Limitations

While this work shows good agreement with the current clinical standard, it is important to acknowledge some of the study’s limitations. The SCPT performed in-clinic was completed using a four-step staircase, which is regarded as an acceptable method for measuring stair climb power, but limited data capture during stair walking events. Our stair walking time-series signals were about 4 s in duration (per task), which limited our training data. Furthermore, our findings are limited in that we only use structured gait as a negative training class and structured stair walking as a positive training class for classifying stair climbing. Including other unstructured, free-living classes would give a more comprehensive understanding of the model’s ability to distinguish stair ascending from other real-world activities. Additionally, it is important to note that the algorithm presented in this paper assumes alternating ascending steps on the staircase and a constant normal-paced walking speed, which may not be feasible for some patient populations. Future work will investigate the ability of the algorithm to generalize to pauses, different walking speeds, and irregular stepping patterns. Lastly, the gold-standard comparator used in assessing the accuracy of our sensor-based stair climbing power estimation is subject to human error. Since a trained technician is responsible for stopping and starting the stopwatch, there can be slight differences between when the participant starts and stops the task versus when the technician initiates and terminates the stopwatch.

### 3.6. Future Work

Future work will include investigating multiclass classification and context detection, which will allow our model to distinguish stair ascending from other physical activities and daily movements, such as standing, running, and swimming. Furthermore, it would be of interest to look at other machine learning models which may be able to enhance classification accuracy. We also intend to apply this algorithm to alternate datasets. Our dataset was gathered from a controlled in-clinic assessment, and it would be of interest to apply this work to separate clinical populations, and observe differences between healthy and disease groups. This creates a more robust classifier which might be better suited for an at-home assessment. Future work will also include evaluation of the classifier in an at-home setting and testing the algorithm’s ability to estimate stair climb power in free-living conditions. While investigating this classifier’s ability in an at home setting, it would be of interest to explore other data collection methods, such as smartphones and smartwatches, which also contain accelerometers and can be positioned around the waist.

## 4. Conclusions

In the findings presented, stair climb power was estimated using a single lumbar-worn accelerometer with strong agreement compared to the clinical assessment. This work demonstrates the feasibility of using a wearable sensor to estimate stair climb power, and highlights the benefits and limitations for using accelerometer and gyroscope sensor modalities for identifying periods of stair walking from level walking, and classifying stair ascending from descending.

The SCPT is an important clinical assessment for measuring exercise capacity in clinical populations, such as cachexia or osteoarthritis [15,48]. The stair climb power package (StairPy) presented in this paper can provide an estimation of stair climb power outside of the clinic, and shows strong agreement with the current clinical standard. If patients are instrumented with a lumbar positioned accelerometer at home, this algorithm potentially provides an end-to-end solution for observing continuous stair climbing performance in free-living conditions. To this end, longer observation of physical function at-home may provide a more comprehensive estimation of true leg muscle functional capacity, as patients may be inclined to over-perform while observed in-clinic due to the Hawthorne effect [49]. We hope the work presented will lead to at-home assessment of stair climb power, and provide enhanced assessment of physical function out of the clinic, as well as a more objective assessment in a clinic setting, not subject to human error.

## Figures and Tables

**Figure 1 sensors-22-06600-f001:**
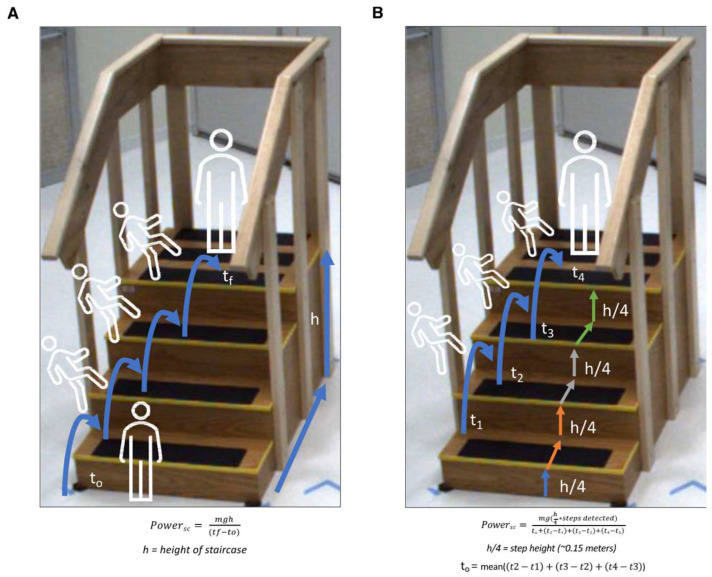
Schematic view of (**A**) Standard (four-step) SCP, where the participant starts with both feet on the ground at time $t_{0}$, which is the time when the clinician says “go”, and finishes at time *t_f_* when both feet touch the top step, and (**B**) StairPy SCP estimation, where the participant starts climbing when the clinician says “go”, and finishes when both feet reach to last step at $t_{4}$. In this method, the each step (*t*_1_, *t*_2_, *t*_3_, *t*_4_) are derived from the StairPy, and the time between the clinican says “go” and the 1st step (*t*_0_) is estimated as the average time between steps (mean ((*t*_2_ − *t*_1_) + (*t*_3_ − *t*_2_) + (*t*_4_ − *t*_3_)). [42].

**Figure 2 sensors-22-06600-f002:**
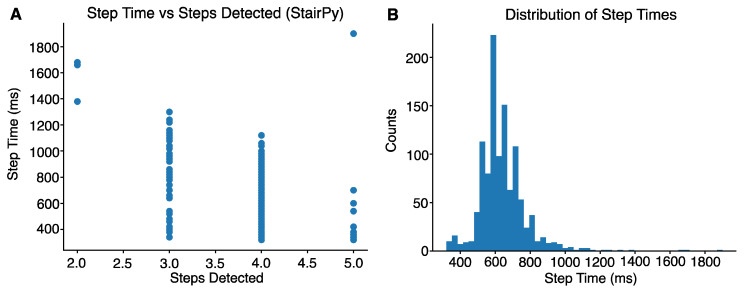
(**A**) Steps greater than 1200 ms (1.2 s) were misidentified as either 2, 3, or 5 total steps and (**B**) Distribution of step times across participants.

**Figure 3 sensors-22-06600-f003:**
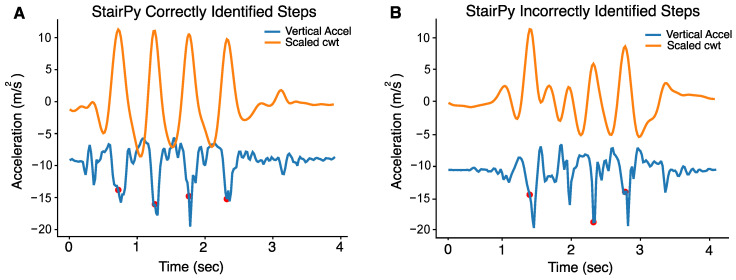
(**A**) Example of correctly identified steps, as four contacts were identified (red) and (**B**) An example of misidentified second step, where only 3 contacts were identified (red) (second step not identified).

**Figure 4 sensors-22-06600-f004:**
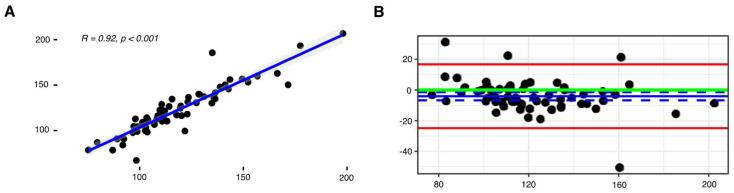
(**A**) Comparison of StairPy derived stair climb power (SCP) and Standard SCP calculation. The stair climb powers were highly correlated (Pearson’s *r* = 0.92, *p* < 0.001. (**B**) Bland Altman (Standard SCP—StairPy SCP) showed minimal mean difference (mean difference = −4.11, blue solid line; LoA = [−24.9, 16.7], green solid line; zero bias reference line (mean difference = 0), red solid lines; corresponding confidence intervals are dashed lines).

**Figure 5 sensors-22-06600-f005:**
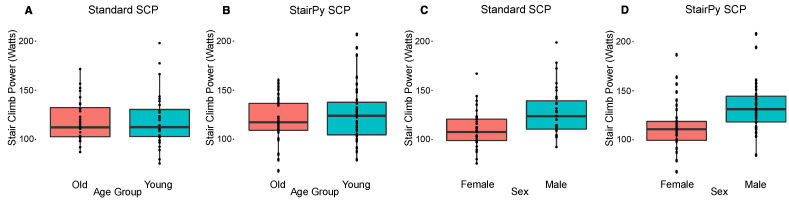
(**A**) The Standard SCP showed no significant differences between younger and older groups (*p*  =  0.96). (**B**) The StairPy derived SCP also showed no significant differences between younger and older groups (*p =* 0.39). (**C**) The Standard SCP showed a significant effect between sex groups (*p*  = 0.002), with males exhibiting higher SCP values and likewise (**D**) the StairPy SCP showed a similar significant difference between sex groups (*p*  =  0.002).

**Figure 6 sensors-22-06600-f006:**
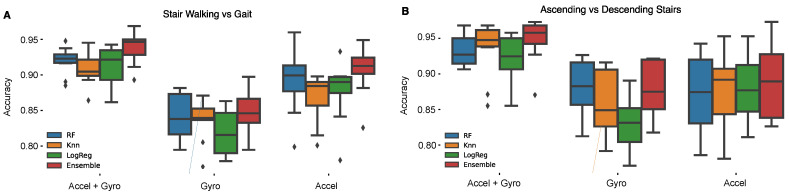
(**A**) Stair walking vs. gait classification was able to achieve the highest performance accuracy using an Ensemble voting classifier model trained on features of the accelerometer and gyroscope signal; however, this accuracy has minimal difference compared to the accelerometer based Ensemble model. (**B**) Stair ascending vs. descending classification was able to achieve the highest performance accuracy using an Ensemble voting classifier model trained on features of the accelerometer and gyroscope signal; however, this accuracy has minimal difference compared to the accelerometer based Ensemble model.

**Figure 7 sensors-22-06600-f007:**
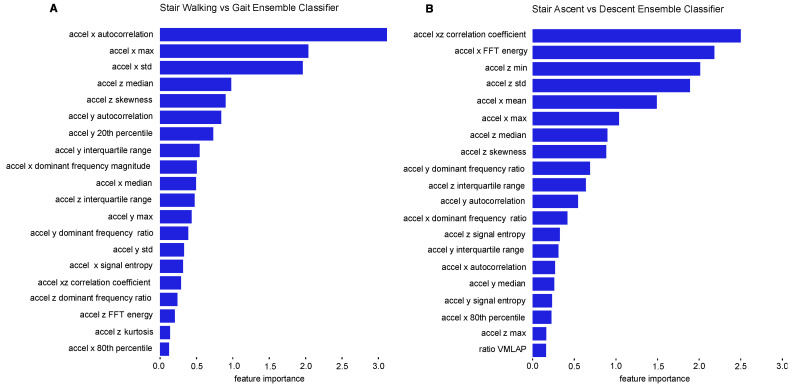
(**A**) Feature importances for accelerometer feature based ensemble stair walking vs. gait classifier. Accelerometer x autocorrelation, accelerometer x maximum, accelerometer x standard deviation, accelerometer z median, accelerometer z skewness, accelerometer y autocorrelation, and accelerometer y 20th percentile were the most predominant features (**B**) Feature importances for ensemble stair ascending vs. descending classifier. Accelerometer xz correlation coefficient, accelerometer x FFT energy, accelerometer z minimum, accelerometer z standard deviation, accelerometer x mean, accelerometer x maximum, and accelerometer z median were the most predominant features.

**Table 1 sensors-22-06600-t001:** Stair walking vs. gait classifier results.

Model	Metric (Mean (std))	Accel	Gyro	Accel + Gyro
RF	accuracy	0.892 (0.044)	0.841 (0.032)	0.920 (0.019)
RF	specificity	0.892 (0.080)	0.869 (0.046)	0.928 (0.049)
RF	sensitivity	0.895 (0.058)	0.826 (0.044)	0.908 (0.035)
Knn	accuracy	0.869 (0.035)	0.838 (0.030)	0.908 (0.023)
Knn	specificity	0.888 (0.065)	0.861 (0.040)	0.948 (0.029)
Knn	sensitivity	0.850 (0.034)	0.807 (0.052)	0.864 (0.046)
LogReg	accuracy	0.878 (0.041)	0.818 (0.031)	0.914 (0.028)
LogReg	specificity	0.882 (0.076)	0.829 (0.045)	0.910 (0.059)
LogReg	sensitivity	0.885 (0.061)	0.805 (0.060)	0.925 (0.030)
Ensemble	accuracy	0.908 (0.035)	0.846 (0.033)	0.938 (0.023)
Ensemble	specificity	0.916 (0.068)	0.875 (0.041)	0.949 (0.038)
Ensemble	sensitivity	0.906 (0.065)	0.824 (0.048)	0.928 (0.034)

**Table 2 sensors-22-06600-t002:** Stair climb ascending vs. descending classifier results.

Model	Metric (Mean (std))	Accel	Gyro	Accel + Gyro
RF	accuracy	0.868 (0.059)	0.880 (0.038)	0.933 (0.024)
RF	specificity	0.884 (0.093)	0.886 (0.054)	0.943 (0.053)
RF	sensitivity	0.866 (0.081)	0.866 (0.083)	0.928 (0.055)
Knn	accuracy	0.877 (0.055)	0.859 (0.047)	0.935 (0.041)
Knn	specificity	0.874 (0.082)	0.854 (0.056)	0.925 (0.073)
Knn	sensitivity	0.893 (0.075)	0.869 (0.068)	0.956 (0.023)
LogReg	accuracy	0.876 (0.049)	0.826 (0.037)	0.921 (0.035)
LogReg	specificity	0.885 (0.068)	0.818 (0.045)	0.917 (0.062)
LogReg	sensitivity	0.872 (0.077)	0.843 (0.063)	0.932 (0.049)
Ensemble	accuracy	0.888 (0.054)	0.878 (0.040)	0.948 (0.032)
Ensemble	specificity	0.901 (0.076)	0.887 (0.046)	0.945 (0.064)
Ensemble	sensitivity	0.886 (0.081)	0.872 (0.071)	0.958 (0.027)

## Data Availability

Upon request, and subject to review, Pfizer will provide the data that support the findings of this study. Subject to certain criteria, conditions and exceptions, Pfizer may also provide access to the related individual de-identified participant data. See https://www.pfizer.com/science/clinical-trials/trial-data-and-results for more information.

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
