# Peer review of "Wearable-Based Stair Climb Power Estimation and Activity Classification"

_sensors, 2022, doi:10.3390/s22176600_

Round 1
Reviewer 1 Report
SUMMARY
This work proposes algorithm to (a) classify stair climbing periods (from walking, and from stairs descending) and (b) estimate "stair climb power" (SCP). Methods are evaluated in data collected in-lab from healthy adults (n=65).
I think this paper tackles the data (continuous signals collected with wearable devices) and problems (activity recognition, developing digital enpoint of a clinical measure) of high interest in recent years. It is well-writted and easy to follow, and has particularly strong Discussion section. I think the paper's usefulness could be increased by discussion on when the method works well versus when it is anticipated to have difficulties is needed in the free-living deployment setup. Some specific comments are included below.
MAJOR
- I am unable to locate stairpy python package. The Link S1 does not work. There is no publically available repository with "stair" in its name within https://github.com/orgs/PfizerRD/repositories . I am wondering if this was intended? It makes me feel GitHub could do with one more layer of repo accessibility, "do not display on repository list but allow access once someone knows url".
- [Materials and Methods > Participants and procedure] I feel more explanation/clarification is needed in the text accompanying Figure 1. First, the Figure already provides the definition of the two SCPs, but the text does not specify them until later subsection "Stair climb power calculation (StairPy)". Second, while the t's for StairPy SCP calculation are clear ("time between alternating initial contacts"), the t's are not clear for clinical SCP (Is "t_0" the moment of lifting one foot? Does "t_f", currently defined as "reach the top of the stairs", refer to initial contact of one foot, or both feet, or somehow else?)
- [Materials and Methods > Participants and procedure] More information is needed regarding the types of data collected by the sensors (State these are 3-axial, what was data collection frequency for each of the sensor type, how the data were extracted/downloaded)
- For stair climbing classification and for SCP estimation, is this assumed that the stairs climbing is continuous to be a well-completed exercise? What happens if there is a slowdown / short break in climbing?
- In potential free-living deployment, would it make more sense to conduct it as a semi-supervised exercise? If not, how do you evnision distinguising between one specific in home stairs, versus any other stairs for the sake of comparing the same measurement in a longitudinal setting?
MINOR
- [Abstract] "This method (...) cannot provide longitudinal observation" -- Is this sentence is accurate? What if I do go to clinic occassionally but regularly to have SCPT performed?
- [Materials and Methods > Participants and procedure] "Participants were equally distributed by sex and age into the two age groups." -- Does this sentence need a fix?
- [Materials and Methods > Participants and procedure] On Figure 1, I wonder if the left-side graphic can be improved, given the current version may suggest a participant makes a 4 stairs "jump" at once, instead of stepping at each stair separately?
- [Materials and Methods > Participants and procedure] On Figure 1, and thorough the text consider using "SCP_{StairPy}" and "SCP_{standard}", or use other naming convenition to distinguish the two definitions (currently both "Power_{sc}").
- [Materials and Methods > Statistical Analyses] I suggest using the established notation of "SCP" instead "power" in the statistical analyses description (e.g., ( "SCP values" and "SCP estimation" instead of "power values" or "power estimation").
Author Response
Dear Reviewer,
We would like to profusely thank you for your helpful and insightful comments, and excellent suggestions! We believe that the modifications and edits implemented in the revised version of the manuscript to address these comments, greatly improved the manuscript. Edits in the manuscript are marked in Red.
Responses to Reviewer 1:
MAJOR
- I am unable to locate stairpy python package. The Link S1 does not work. There is no publically available repository with "stair" in its name within https://github.com/orgs/PfizerRD/repositories . I am wondering if this was intended? It makes me feel GitHub could do with one more layer of repo accessibility, "do not display on repository list but allow access once someone knows url".
The repository is currently in a Pfizer internal repository and not publicly available, however, we are awaiting approval for a secondary repository which will be publicly available.
- [Materials and Methods > Participants and procedure] I feel more explanation/clarification is needed in the text accompanying Figure 1. First, the Figure already provides the definition of the two SCPs, but the text does not specify them until later subsection "Stair climb power calculation (StairPy)". Second, while the t's for StairPy SCP calculation are clear ("time between alternating initial contacts"), the t's are not clear for clinical SCP (Is "t_0" the moment of lifting one foot? Does "t_f", currently defined as "reach the top of the stairs", refer to initial contact of one foot, or both feet, or somehow else?)
Figure 1 has been moved to a more appropriate section of the paper (Materials and Methods/Stair climb power calculation (StairPy) – page 6), and we have further elaborated on the description for Figure 1. In the description, we have explicitly described the times t0 and tf for the Clinical SCP, and clarified that the time tf is when both feet reach the top step.
- [Materials and Methods > Participants and procedure] More information is needed regarding the types of data collected by the sensors (State these are 3-axial, what was data collection frequency for each of the sensor type, how the data were extracted/downloaded)
We have added notes about the sampling frequency (128 Hz) and the method of data extraction, clarifying that data was locally download using Mobility Lab software, and exported as deidentified .H5 files.
- For stair climbing classification and for SCP estimation, is this assumed that the stairs climbing is continuous to be a well- completed exercise? What happens if there is a slowdown / short break in climbing?
The stair climb activity performed in-clinic was done continuously at the participant’s normal pace, with a 30sec break in between in order to reduce fatigue. Future work will explore the application of this algorithm in an at-home environment with continuous monitoring, which will allow us to evaluate whether the algorithm can generalize to different walking speeds or pauses. We have now added this to the Discussion/Future Work section.
- In potential free-living deployment, would it make more sense to conduct it as a semi-supervised exercise? If not, how do you evnision distinguising between one specific in home stairs, versus any other stairs for the sake of comparing the same measurement in a longitudinal setting?
A semi-supervised approach is intriguing and certainly feasible; however, this algorithm is designed to be used for passive monitoring. We intend to monitor participants in their at-home environment, without encouraging activity. We believe passive monitoring is the best approach to receiving complete and non-biased activity data while at-home and removing any elements of the Hawthorn Effect.
MINOR
- [Abstract] "This method (...) cannot provide longitudinal observation" -- Is this sentence is accurate? What if I do go to clinic occasionally but regularly to have SCPT performed?
We have changed longitudinal to continuous remote monitoring in the Abstract: “This method is subject to human error and cannot provide continuous remote monitoring”. The goal of this algorithm is at-home observation. We want to offer an alternative to in-clinic measures which could be applied to decentralized clinical studies.
- [Materials and Methods > Participants and procedure] "Participants were equally distributed by sex and age into the two age groups." -- Does this sentence need a fix?
This sentence has been reformatted to be clearer to readers, indicating that the number of male and female participants was equal in both the older and younger cohorts: “Participants were equally distributed by sex in the older and younger age groups”.
- [Materials and Methods > Participants and procedure] On Figure 1, I wonder if the left-side graphic can be improved, given the current version may suggest a participant makes a 4 stairs "jump" at once, instead of stepping at each stair separately?
We have adjusted figure 1 to clarify that participants are walking up the staircase and not“jumping”.
- [Materials and Methods > Participants and procedure] On Figure 1, and thorough the text consider using "SCP_{StairPy}" and "SCP_{standard}", or use other naming convenition to distinguish the two definitions (currently both "Power_{sc}").
We have incorporated this adjusted naming convention and have applied it throughout the manuscript.
- [Materials and Methods > Statistical Analyses] I suggest using the established notation of "SCP" instead "power" in the statistical analyses description (e.g., ( "SCP values" and "SCP estimation" instead of "power values" or "power estimation").
We have incorporated this change to the Materials and Methods/Statistical Analysis section of the paper (page 8), and believe it will improve the interpretability of the manuscript.
Reviewer 2 Report
This paper presents a technique to estimate stair climb power of human using lumbar-positioned accelerometer and gyroscope. The proposed technique extracts a few very simple handcrafted features rom the windowed segmented sensory data i.e., mean, variance, root mean square and etc. to distinguish stair walking. For the stair climbing classification, a comparative study of three well-known machine learning algorithms (k-NN, RF and LogReg) is presented.
In my point of view, the following concerns should be addressed:
· The proposed methodology is lacking with novelty. The authors employed some standard set of features (i.e., mean, variance, root mean square) and machine learning tools (i.e., k-NN, RF and LogReg) to classify the stairs walking which have been used in several paper from last many years. The scientific contribution must be stated clearly.
· Rather than using very simple feature encoding technique; I would like to see the evaluation of 16 handcrafted features proposed in [1].
[1] F. Amjad, M.H. Khan, M.A. Nisar, M.S. Farid, M. Grzegorzek, “A comparative study of feature selection approaches for human activity recognition using multimodal sensory data, ” Sensors, vol. 21, 2021
· The authors should publicly available the collected dataset to the research community and this may be treated as a strong contribution.
· Wearing a set of IMU on human-body can minimize the actual movement of the subject’ to some extent. Why the authors did not explore the same sensory data from the devices of daily usage e.g., Mobile device. For reference see [2].
[2] Köping, Lukas, Kimiaki Shirahama, and Marcin Grzegorzek. "A general framework for sensor-based human activity recognition." Computers in biology and medicine 95 (2018): 248-260.
· The experimental evaluation is very limited. The computed results must compare with state-of-the-art techniques.
· The author should improve the writing style of the paper. It shall look like a research article rather than an experimental report.
Author Response
Dear Reviewer,
We would like to profusely thank you for your helpful and insightful comments, and excellent suggestions! We believe that the modifications and edits implemented in the revised version of the manuscript to address these comments, greatly improved the manuscript. Edits in the manuscript are marked in Red.
Responses to Reviewer 2:
The proposed methodology is lacking with novelty. The authors employed some standard set of features (i.e., mean, variance, root mean square) and machine learning tools (i.e., k-NN, RF and LogReg) to classify the stairs walking which have been used in several paper from last many years. The scientific contribution must be stated clearly.
We have added a statement to the Discussion/Main Findings section (page 12) outlining the specific scientific contributions: 1. An open-source package for estimating stair climb power from lumbar positioned accelerometer 2. Comparison of different sensor combinations for classification of stair climb power. 3. Demonstrating the ability of an ensemble classifier to identify stair walking and stair ascent with >89% accuracy.
Rather than using very simple feature encoding technique; I would like to see the evaluation of 16 handcrafted features proposed in [1].
[1] F. Amjad, M.H. Khan, M.A. Nisar, M.S. Farid, M. Grzegorzek, “A comparative study of feature selection approaches for human activity recognition using multimodal sensory data, ” Sensors, vol. 21, 2021
We have added most of the suggested hand crafted features noted in Amjad et al. to our feature set. We noted a slight increase in accuracy (~2% across models) and credit this model improvement to the reviewer’s suggested changes.
The authors should publicly available the collected dataset to the research community and this may be treated as a strong contribution.
According to Pfizer policy, interested parties are welcome to submit a request to the author for the data, which will be reviewed and approved by Pfizer. Please see Data Availability Statement (page 20).
Wearing a set of IMU on human-body can minimize the actual movement of the subject’ to some extent. Why the authors did not explore the same sensory data from the devices of daily usage e.g., Mobile device. For reference see [2].
Data was collected from a Pfizer Internal study, where the device selection included gold standard devices and the wearables, we had previous experience with, not mobile technologies like smartphones.
The study aims included the aims of: 1. Using digital measurements to distinguish between old/young cohorts, 2. Assess the feasibility of instrumented functional endpoints in-clinic and at-home in old/young cohorts, 3. Assessment of wearable devices for usability and agreement with gold standard devices, 4. Assessment of continuous at-home measurement, 5. Assess agreement of uninstrumented and instrumented measurements of physical performance in clinic, 6. Assess the test-retest reliability and reproducibility of uninstrumented and instrumented functional endpoints, 7. Assess effect of age and sex on instrumented and uninstrumented endpoints, 8. Generate instrumented and uninstrumented control data for design of future cachexia studies.
Stair Climb Power estimation was not a main objective in this study, rather an exploratory project which we were able to develop with data collected in the aforementioned study.
I have added a note that future work will include applying this technique to data collected using mobile technologies like smartphones and smart watches.
[2] Köping, Lukas, Kimiaki Shirahama, and Marcin Grzegorzek. "A general framework for sensor-based human activity recognition." Computers in biology and medicine 95 (2018): 248-260.
The experimental evaluation is very limited. The computed results must compare with state-of-the-art techniques.
Deep learning is one of the state-of-the-art methods that is currently used in the literature. However, our sample size and amount of data is not enough for this type of data-hungry techniques. Instead, we have added an ensemble classification method, leveraging the Logisitic Regression, Random Forest, and Knn classification models in a Voting Classifier. This method is novel and has not been previously explored in the stair climb power field. Using this method we were able to achieve >89% accuracy for both models.
The author should improve the writing style of the paper. It shall look like a research article rather than an experimental report.
We have substantially edited the manuscript and restructured some sections to make it look more like a research article.
Reviewer 3 Report
1) The paper can be accepted if the authors support their findings using more advanced machine learning methods from the relevant literature and not just KNN or Random Forest.
2) The paper should be formatted according to the style of MDPI
Author Response
Dear Reviewer,
We would like to profusely thank you for your helpful and insightful comments, and excellent suggestions! We believe that the modifications and edits implemented in the revised version of the manuscript to address these comments, greatly improved the manuscript. Edits in the manuscript are marked in Red.
Responses to Reviewer 3:
1) The paper can be accepted if the authors support their findings using more advanced machine learning methods from the relevant literature and not just KNN or Random Forest.
We have added a more advanced machine learning method (ensemble classification, which leverages the three models previously trained in a voting classifier. This method is novel and has not been previously explored in the field. Using this method, we were able to achieve >89% accuracy for both models.
2) The paper should be formatted according to the style of MDPI
We have substantially edited the manuscript and restructured some sections in accordance with the suggested framework of MDPI.
Reviewer 4 Report
The manuscript address a very interesting and clinically relevant topic on technology-based monitoring of daily-life physical performance (i.e. stair climb power) in older adults. It presents an algorithm for detecting stair climbing periods and estimating stair climb power from an accelerometer. The manuscript is very-well written, gives a clear study rational, provides adequate information for reproducibility, and presented the results very clearly. I enjoyed reading the manuscript, and do have only very minor comments.
Abstract:
- Please introduce the abbreviation ICC.
- Please do not provide the p-value in p<2.2x10-16, but rather p < 0.001 (also across the entire manuscript)
Methods: What were the in- and exclusion criteria for study participation?
Author Response
Dear Reviewer,
We would like to profusely thank you for your helpful and insightful comments, and excellent suggestions! We believe that the modifications and edits implemented in the revised version of the manuscript to address these comments, greatly improved the manuscript. Edits in the manuscript are marked in Red.
Responses to Reviewer 4:
Abstract:
Please introduce the abbreviation ICC.
We have added language to introduce ICC in the Materials and Methods/Statistical Analyses section (page 8).
- Please do not provide the p-value in p<2.2x10-16, but rather p < 0.001 (also across the entire manuscript)
We have incorporated the suggested changes to the p-values in the paper (p < 0.001).
Methods: What were the in- and exclusion criteria for study participation?
We have incorporated the inclusion and exclusion criteria to the manuscript, in the Materials and Methods/Participants and Procedure section (pages 3-4).
Round 2
Reviewer 2 Report
Since the authors employed a few feature encoding techniques from the following paper, it should be ethically cited in their manuscript.
F. Amjad, M.H. Khan, M.A. Nisar, M.S. Farid, M. Grzegorzek, “A comparative study of feature selection approaches for human activity recognition using multimodal sensory data, ” Sensors, vol. 21, 2021
Author Response
Dear Reviewer,
We would like to profusely thank you for your helpful and insightful comments, and excellent suggestions! We believe that the modifications and edits implemented in the revised version of the manuscript to address these comments, greatly improved the manuscript. Edits in the manuscript are marked in Red.
Responses to Reviewer 2:
Since the authors employed a few feature encoding techniques from the following paper, it should be ethically cited in their manuscript.
- Amjad, M.H. Khan, M.A. Nisar, M.S. Farid, M. Grzegorzek, “A comparative study of feature selection approaches for human activity recognition using multimodal sensory data, ” Sensors, vol. 21, 2021
This paper was already ethically cited in the updated manuscript we submitted on Aug 23rd and can be found in the citation list:
- Amjad, F., et al., A Comparative Study of Feature Selection Approaches for Human Activity Recognition Using Multimodal Sensory Data. Sensors (Basel), 2021. 21(7).
We have also submitted the revised manuscript again and highlighted this citation in “red” for your convenience.
Sincerely,
Dimitrios Psaltos